# Capturing the inherent structural dynamics of the HIV-1 envelope glycoprotein fusion peptide

Sonu Kumar [1,2,3], Anita Sarkar [1,2,3], Pavel Pugach[4], Rogier W. Sanders[4,5], John P. Moore [4], Andrew B. Ward [1,2,3] & Ian A. Wilson [1,2,3,6]

The N-terminal fusion peptide (FP) of the human immunodeficiency virus (HIV)-1 envelope glycoprotein (Env) gp41 subunit plays a critical role in cell entry. However, capturing the structural flexibility in the unbound FP is challenging in the native Env trimer. Here, FP conformational isomerism is observed in two crystal structures of a soluble clade B transmitted/founder virus B41 SOSIP.664 Env with broadly neutralizing antibodies (bNAbs) PGT124 and 35O22 to aid in crystallization and that are not specific for binding to the FP. Large rearrangements in the FP and fusion peptide proximal region occur around M530, which remains anchored in the tryptophan clasp (gp41 W623, W628, W631) in the B41 Env prefusion state. Further, we redesigned the FP at position 518 to reinstate the bNAb VRC34.01 epitope. These findings provide further structural evidence for the dynamic nature of the FP and how a bNAb epitope can be restored during vaccine design.

[1] Department of Integrative Structural and Computational Biology, The Scripps Research Institute, La Jolla, CA 92037, USA. [2] IAVI Neutralizing Antibody Center, The Scripps Research Institute, La Jolla, CA 92037, USA. [3] Center for HIV/AIDS Vaccine Immunology and Immunogen Discovery, The Scripps Research Institute, La Jolla, CA 92037, USA. [4] Department of Microbiology and Immunology, Weill Medical College of Cornell University, New York, NY 10065, USA. [5] Department of Medical Microbiology, Academic Medical Center, University of Amsterdam, 1105 AZ Amsterdam, The Netherlands. [6] Skaggs Institute for Chemical Biology, The Scripps Research Institute, La Jolla, CA 92037, USA. Correspondence and requests for materials should be addressed to I.A.W. (email: wilson@scripps.edu)

The metastable nature of cleaved, fusion-competent human immunodeficiency virus (HIV)-1 envelope glycoprotein (Env) controls the critical structural rearrangements that are required for fusion between the viral and host cell membranes after sequential binding to the CD4 receptor and the CXCR4 or CCR5 co-receptor[1]. The gp120 subunits of Env house the receptor and coreceptor-binding sites, while gp41 contains the fusion machinery[2,3]. The culmination of this cell entry process is orchestrated by the fusion peptide (FP) in the gp41 subunit through its insertion into the host cell membrane. The hydrophobic FP at the N-terminus of gp41 becomes available for fusion activity after proteolytic cleavage of gp160 into gp120 and gp41 by a host cell serine protease of the Furin family. After cleavage, Env adopts a metastable prefusion conformation. Upon receptor and co-receptor binding, steric constraints are released to allow the three N-terminal and C-terminal heptad repeats of gp41 in the Env trimer to transition to the highly stable six-helix bundle conformation that brings the viral and host membranes into close enough proximity for fusion[3,4]. The FP is therefore directly involved in the transition from the pre-fusion state to the intermediate and post-fusion states. This intrinsic dynamic nature of Env is critical for controlling and timing the molecular recognition events that lead to cell entry and infection, but creates challenges for vaccine design[5–8].

To construct a soluble, cleaved, native-like gp140 trimer as a potential vaccine immunogen, HIV-1 Env was first stabilized with a disulfide bond between gp120 and gp41 and an I559P mutation in gp41; a more efficient cleavage site and truncation at residue 664 created the SOSIP.664 design[7–12]. These major advances enabled determination of the x-ray and cryo-EM structures of the closed, prefusion native-like HIV-1 Env[13–15]. Other vaccine immunogen platforms ensued and were based on cleavage-independent designs that either included the I559P mutation in single-chain gp140 (sc-gp140)[16] or native flexibly linked gp140 (NFL-gp140)[17], or had a completely modified HR1N (UFO)[18]. Both cleaved (SOSIP) and cleavage-independent (NFL)[19,20] structures show that the HR1N, FP (residues 512–527), and FPPR (residues 528–540) regions are highly flexible[21] and may benefit from further gp41 stabilization. Therefore, there is still room for further improvement in design of stable HIV-1 Env immunogens that include the FP region and mutations to stabilize the prefusion structure[7,8,18,20,22–27].

Presently, out of several soluble SOSIP.664 designs, BG505 SOSIP.664 from clade A[10], B41 SOSIP.664 from clade B[28], and CZA97 SOSIP.664 from clade C[29], have been shown to induce strong autologous NAbs against neutralization-resistant (tier 2) viruses[8,30]. These soluble trimers are close mimics of the native trimer on the surface of virions and are promising immunogens for a nAb-eliciting HIV-1 vaccine[8,23,29,30]. Both BG505 and B41 SOSIPs have the propensity to bind to almost all broadly neutralizing antibodies (bNAbs). However, BG505 SOSIP.664 ($T_m = \sim 68\,^{\circ}\mathrm{C}$)[10] is more thermostable than B41 SOSIP.664 ($T_m = \sim 57\,^{\circ}\mathrm{C}$)[28], a property that could be a reflection of the greater conformational flexibility of B41 SOSIP.664 as visualized by negative-stain electron microscopy (NS-EM)[28]. Recently, good agreement on Env compactness was reported between unbound and antibody-bound Envs in solution and on the surface of virions, where structural homogeneity was observed at the trimer apex but with some variation in conformation at the base of the Envs[21]. Conformational variation in Env is, therefore, not only induced by binding of receptors/co-receptors to Env, but also by flickering motions within the sub-domains. A dynamic equilibrium is maintained between different conformations, including partial opening/breathing at the trimer apex and some movement in gp41[21,31,32]. The relaxation of V1/V2 and V3, termed breathing, at the trimer apex facilitates PGT145 binding[33]. NS-

EM and cryo-EM have shown that the unliganded B41 SOSIP.664 trimer exhibits two-subpopulations: closed (prefusion: more native-like) and intermediate (prefusion: equilibrium between closed and more open) conformations. The sequential binding of the soluble CD4 receptor and Fab 17b induces open Env conformations[28,31].

Analyses of the CD4-bound open prefusion conformations of the B41 SOSIP.664 trimer and the conformational rearrangements in the FP[31] have drawn attention towards the inherent flexibility of FP. Recently, the FP was identified to be a target for bNAbs[34–37]. However, little is understood about the impact of FP flexibility on the prefusion dynamics and compactness of Env. To investigate FP flexibility and provide insights into conformational rearrangements associated with membrane fusion and also possibly virus neutralization, we determined two crystal structures of B41 SOSIP.664 trimers that highlight different conformations of the FP and FP proximal region (FPPR) in gp41. Based on the structural findings, we restored the epitope of VRC34.01, a bNAb targeting the FP. We thus have mapped the structural flexibility in this FP region when the epitope is unliganded and not bound by FP-specific bnAbs. These findings help lay a foundation for further structure-guided modifications to improve HIV-1 Env stability, as well as FP-epitope restoration, if required, for bnAb binding and neutralization.

## Results

**Closed prefusion structure of B41 SOSIP.664.** We chose the soluble SOSIP.664 Env of the T/F virus B41 for investigation of structural flexibility within the Env trimer sub-domains, as it was reported to contain mixed conformational populations[28]. Partially open conformations, distinct from the CD4-induced fully-open state, were attributed to loop and/or domain movements that equilibrate between closed and partially open states on HIV-1 Envs[31,32,38–41]. We found that B41 SOSIP.664 trimers expressed in either HEK293S cells (thermal denaturation midpoint, $T_m = 57.2\,^{\circ}\mathrm{C}$) or HEK293F cells ($T_m = 57.6\,^{\circ}\mathrm{C}$) cells are stable (Supplementary Figure 1A), but less so than BG505 SOSIP.664[10]. For crystallographic studies, the B41 SOSIP.664 trimers were further stabilized by binding to Fabs PGT124 and 35O22, which increase the melting temperatures by ~5 °C (to 62.8 °C) and an additional ~1.5 °C (to 64.5 °C), respectively (Supplementary Fig. 1b). After complex formation, the unprotected glycans were trimmed with EndoH glycosidase and the trimer-Fab complexes were then purified using SEC (Supplementary Fig. 1d). The trimer-Fab complex crystallized in two different conditions, and x-ray structures were determined at 3.50 Å and at 3.80 Å resolutions in cubic (P23) and hexagonal (P6₃) crystal lattices (Fig. 1a, b, Supplementary Fig. 1e and f) with data complete to ~100% in the highest resolution shell (Supplementary Table 1). PGT124 recognizes the N332 glycan and the ³²⁴GDIR³²⁷ motif at the V3 base on B41 SOSIP.664 trimers with an angle of approach similar to ones seen with other trimers[42]. We detect conserved interactions (N325₍gp120₎ to S30 in CDRL3, S93 in CDRL1 and Y100b in CDRH3) despite a D325N mutation in the ³²⁴GDIR³²⁷ motif compared to BG505 (Supplementary Fig. 2). The Asp/Asn duo appears at an 80%/17% frequency at position 325 across 5451 aligned HIV-1 Env sequences (Los Alamos HIV database). B41 SOSIP.664 (3.50 Å) adopts overall protomer/trimer conformations, similar to previously determined trimer structures from other isolates/clades[25,43–45], with Cα root mean square deviations (r.m.s.d) ranging between 0.6–1.0 Å (protomer) and 1.0–1.3 Å (trimer) (Fig. 1c). Thus, despite no significant gp120 conformational differences, the same B41 SOSIP.664 complex could be crystallized in two different crystal lattices (Supplementary Fig. 3). However, gp41 exhibits substantial conformational

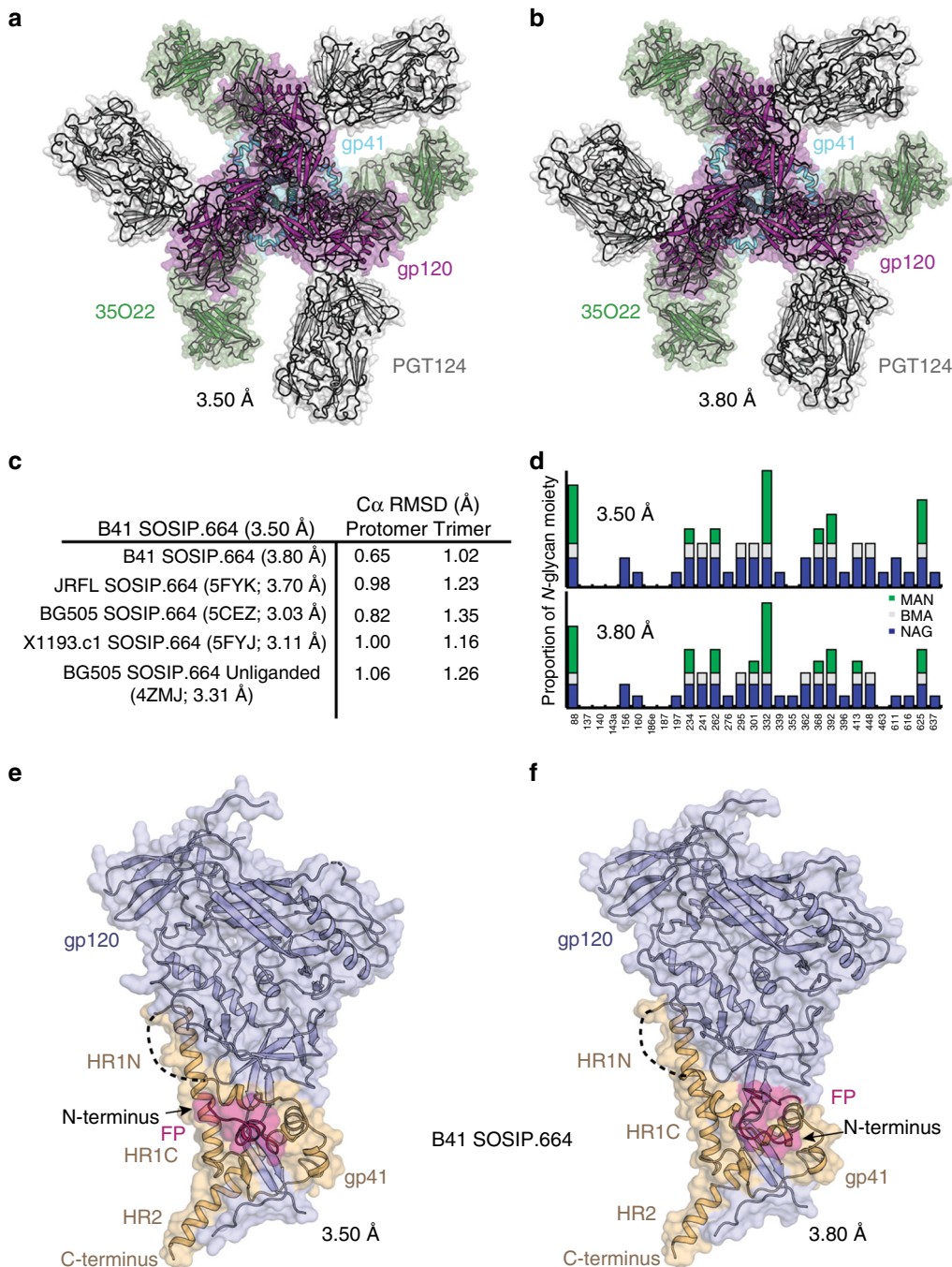

**Fig. 1** Crystal structures of closed prefusion structure of B41 SOSIP.664 Env trimer. **a**, **b** Views of the Fab complex down the trimer axis with gp120 in pink and gp41 in cyan. The cartoon representation is overlaid with a transparent molecular surface. Fabs PGT124 (dark gray) and 35O22 (green) are from the 3.50 and 3.80 Å structures in space groups P23 (**a**) and P6$_3$ (**b**), respectively. **c** Comparison of the protomer/trimer backbones from crystal structures of trimers of clade A (BG505 SOSIP.664, bound and unbound), B (JRFL SOSIP.664), G (X1193.c1 SOSIP.664), and the new B41 SOSIP.664 structure at 3.50 Å. **d** Glycosylation observed in the electron density maps in the two different B41 SOSIP.664 crystal structures. The blue, gray, and green bars along the Y-axis represent the number of N-glycan moieties of N-acetylglucosamine (NAG/GlcNAc) (maximum possible, two), β-D-mannopyranose (BMA) (maximum, one), and α-D-mannopyranose (MAN) (maximum, eight), respectively. The X-axis represents N-glycosylation sites on B41 SOSIP.664. **e**, **f** Side views of B41 SOSIP.664 protomer (gp120: light blue, gp41: light orange) in cartoon representation, overlaid with a transparent molecular surface showing two conformational states of the free N-terminus FP (pink) going away from (3.50 Å) and towards (3.80 Å) the C-terminus (i.e. proximal to the membrane)

heterogeneity at its N-terminus (Fig. 1e, f), although the HR1N regions in both crystal lattices are disordered. These features may explain in part the ~10 °C lower thermal stability compared to BG505 SOSIP.664 ($T_m = 68.1$ °C)[10]. Mutations in FPPR (T538F) and HR1N (I548F) of B41 trimer showed improvement in thermal stability[46]. We observed electron density at 24 of the 29

potential N-glycosylation sites (PNGS) present on B41 SOSIP.664 trimers (Fig. 1d)[47], whereas the other five sites are mainly located in disordered loops. The N289 glycan is 72% conserved between HIV-1 isolates, but is not present in B41 (n.b. the sequence around 289 is $^{289}$NEA$^{291}$, which does not code for a glycan). The glycan hole on the wild-type (WT) B41SOSIP.664 trimer is

similar to that on BG505, which lacks both N241 and N289[29,48]. N-glycosylation sites at positions 241, 295, 339, and 448 surround the B41$_{N289}$ glycan hole (Supplementary Fig. 4). Elicited antibodies targeting this immunogenic glycan hole, common to both BG505 and B41 viruses, could possibly be exploited for heterologous cross-neutralization in sequential boosting strategies with HIV-1 trimers[29].

**Flexibility in FP and its proximal region.** To gauge the flexibility of the hydrophobic FP in the prefusion state, we compared the gp41 sub-domains in both crystal structures comprising of B41 SOSIP.664 bound to bNAbs PGT124 and 35O22 (Fig. 2). The FP, which is fully resolved to A512 at the N-terminus in the P23 lattice, is in a conformation that points away from the C-terminus of gp120 (Fig. 2a) and is sandwiched between HR2 of the neighboring protomer in the trimer and the light-chain (LC) variable region (CDRL1 and LFR3) of 35O22 (from another ternary complex in the crystal lattice) (Supplementary Fig. 5a). The FP turns upwards and is stabilized through polar (H641 and T644) and hydrophobic (L645 and V648) interactions with HR2 of the neighboring protomer in the same Env trimer (Supplementary Fig. 5b). In contrast, the fully resolved FP in B41PGT12$_{4+35O22}$ complex in the hexagonal lattice P6$_3$ (Fig. 2b) points toward the C-terminus of gp120. Strikingly, on superimposing the two crystal structures, we observe large rearrangements (up to ~11.6 Å) in the FPPR without altering the M530 position in the tryptophan clasp (W623, W628, W631) that anchors the gp41 sub-domain in the prefusion conformation (Fig. 2c and Supplementary Fig. 6). Despite rearrangements in FP and its proximal region (FPPR), the overall Env conformation in both structures remain unchanged. Comparing the B41 SOSIP.664 structure in P6$_3$ to structures of PGT151-bound WT JRFL ΔCT (PDB 5FUU)[44], VRC34.01-bound BG505 SOSIP.664 (PDB 5I8H)[35,44], and vFP16-bound and vFP20-bound BG505 DS-SOSIP (PDB 6CDI, 6CDE)[36], we observe different

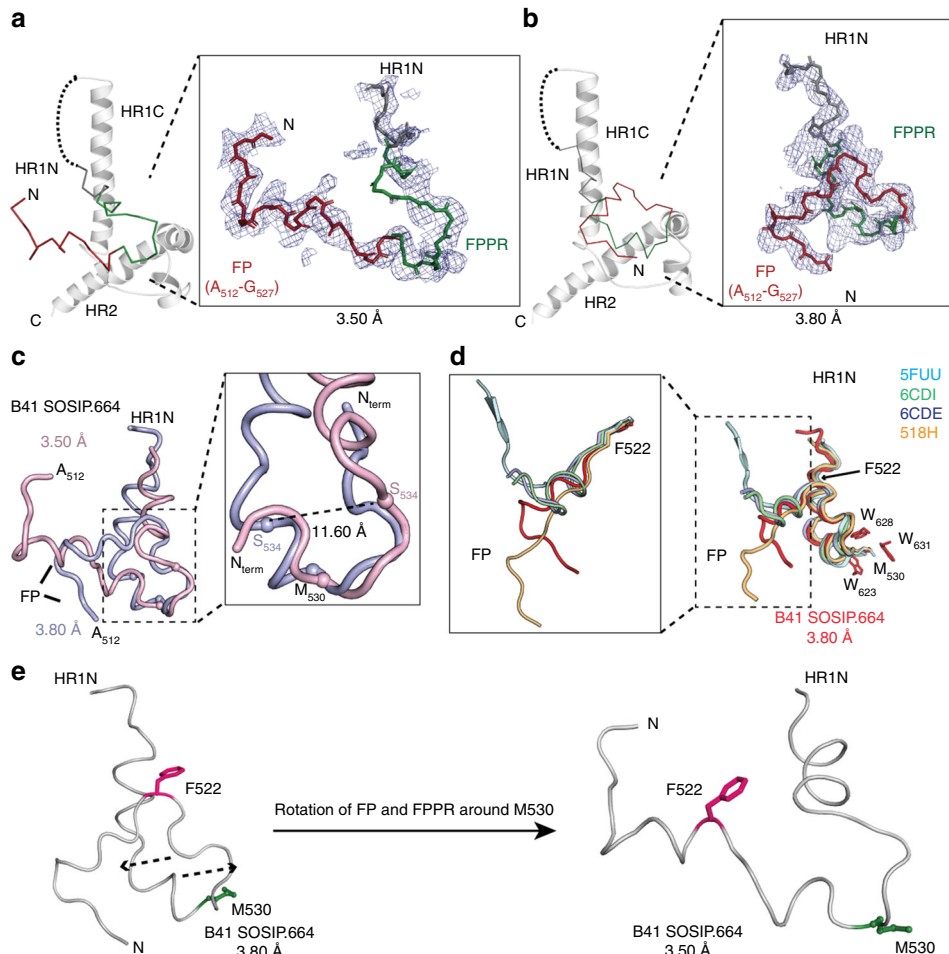

**Fig. 2** Different conformational states of the fusion peptide in B41 SOSIP.664. **a** The FP (red) of gp41 at 3.50 Å points away from HR2. Inset shows the 2Fo–Fc electron density composite omit maps (1.0σ) for the fusion peptide (residues 512–527, red), and FPPR (residues 528–540, green). **b** The FP from the 3.80 Å structure extends towards HR2. Inset shows the 2Fo–Fc electron density map (1.0σ) for the FP and FPPR. **c** Overlay of FP and FPPR regions of gp41 from the two crystal structures (3.50 Å: light pink, 3.80 Å: light blue). Inset illustrates the large rearrangement of FPPR and major Cα displacement for example of S534 (~11.6 Å) in B41 SOSIP.664 following the same resolution-based color-code. **d** Different conformational states of FP illustrated in the overlay of the FP, FPPR, and HR1N regions of PGT151-bound WT JRFL ΔCT (PDB ID: 5FUU; cyan), vFP16-bound BG505 DS-SOSIP (PDB ID: 6CDI; green), vFP20-bound BG505 DS-SOSIP (PDB ID: 6CDE; blue), and VRC34.01-bound BG505 SOSIP.664 (PDB ID: 5I8H; orange), recognized by the respective bNAbs. Inset shows the structural plasticity of the FP around F522. **e** F522 in FP (pink) of B41 SOSIP.664 in the 3.80 Å structure is tucked inside the gp120/gp41 interface and acts as a pivot for the remaining FP residues. Another conformational state of FP is captured in the 3.50 Å structure where F522 exits its hydrophobic-binding pocket, causing the FP and FPPR to rotate in opposite directions without affecting the location and conformation of M530 (green), which acts an anchor in the gp41 prefusion state

conformations that suggests the FP is dynamic and flexible (Fig. 2d). In the FP-antibody bound or unbound trimer structures, FPPR and HR1N adopt overall similar conformations up to the hydrophobic F522 residue, which appears to be the pivot point for the FP structural plasticity[37] and is buried inside a pocket formed at the gp120/gp41 interface (inset, Fig. 2d). In the P23 structure, F522 exits the interface pocket and the FP and FPPR regions rotate with M530 acting as the anchor (Fig. 2e). This altered conformation due to loss of the F522 interface interaction does not otherwise perturb the gp41 prefusion state. Such flexibility in the FP may aid the Env trimer in accessing the conformations observed in the pre-fusion state, antibody-bound states (e.g. PGT151), and the transition to intermediate states. These include the fully closed native-like state and those in equilibrium between the closed and more open conformations observed by NS-EM in different strains and clades[28]. It has been reported that optimizing FPPR (L543N) and HR1N in B41 SOSIPv3.2 improves PGT145 binding and stability, respectively[23].

The current arsenal of crystal and cryo-EM structures illustrate that the FP has a dynamic range of conformations, independent of crystal contact formation, which facilitate bNAb (VRC34.01, ACS202, vFP16, vFP20, and PGT151) engagement from various angles of approach. These results illustrate two representative conformational states where the FP is fully solvent-exposed and accessible to FP-specific antibodies. This FP and FPPR conformational flexibility may be contributing factors that affect the stability[46] and expression of B41 versus BG505 SOSIP trimers, despite eliciting comparable titers of autologous antibody responses[8]. However, only a single conformation of FP in the cleavage-independent (NFL) trimers[17] has been observed to date that points towards the C-terminus[18,19]. The N-terminus of the FP is not free on the cleavage-independent sc-gp140[16] and NFL trimers, as the furin-cleavage site has been deleted. This design feature limits the presentation and/or accessibility of the epitopes for the various FP-directed antibodies. As a result, the FP-directed human bNAbs VRC34.01, ACS202, and PGT151 bind poorly, with high off-rates, to cleavage-independent trimers[34,35]. Further introducing an enterokinase cleavage site, for post-expression enzymatic cleavage, upstream of the FP in NFL improved binding for VRC34.01 and PGT151[49], highlighting the requirement for a charged and free N-terminus for this epitope.

**Restoration of the VRC34.01 epitope on the B41**. To test our hypothesis that the amino-acid sequence and register of certain FP residues have an important role in antibody binding, we analyzed the FP sequence conservation in HIV-1 group M, including recombinants, and compared the sequences to BG505 SOSIP.664 (Fig. 3a). We found that positions 515, 518, and 519 are relatively poorly conserved in the B41 sequence compared to group M as a whole (Supplementary Fig. 7a). Thus, residue L515 on B41 is 41% conserved whereas I515 on BG505 is 53% conserved, showing that both Leu and Ile can be tolerated at FP position 515. However, an I515L mutation to BG505-Env pseudovirus did not affect VRC34.01 neutralization[35]. Much less conservation is seen for B41 residues F518 (~1.5%) and I519 (~13.0%), implying that antibodies with epitopes that include these adjacent residues may be particularly susceptible to antigenic variation in the FP. Although we captured a FP conformation that highly resembles the VRC34.01-bound state on BG505 (PDB 5I8H) (Fig. 2d), this bNAb did not bind to B41 SOSIP.664 trimers (Fig. 3c). To restore the VRC34.01 epitope to the B41 trimer, we aligned the FPs (residues 512–527) of BG505 and B41 and then genetically engineered insertion of Val at position 518 and deletion of Ile at position 519 in B41 SOSIP.664 trimer. In the modified sequence, Phe518 then moved to position

519. Thus, the modified B41 FP sequence is aligned with BG505 except for position 515, which is Leu in B41 but Ile in BG505 (Fig. 3b). We used isothermal titration calorimetry (ITC) to test the binding of VRC34.01 to the FP mutant, designated B41mut1 SOSIP.664, and observed a large increase in affinity compared to the parental B41 SOSIP.664 trimer (Fig. 3c). We conclude that residue 518 is critical for the binding of VRC34.01 to the FP of the B41 trimer.

To understand the underlying structural basis for this epitope restoration, we used biolayer interferometry (BLI) to measure the binding of the VRC34.01 Fab to a C-terminally His-tagged B41mut1 FP (residues 512–521) (Fig. 3d), and then determined its x-ray structure at 1.98 Å resolution (Fig. 3e and Supplementary Table 2). Clear electron density was observed for the entire B41mut1 FP, including the His$_6$-tag (Supplementary Fig. 7a). The paratope, comprising of four shallow cavities formed by all CDR loops, except L2, engages this restored FP (Supplementary Fig. 7b) similar to that observed for VRC34.01 bound to the FP of BG505 SOSIP.664 (Supplementary Fig. 7c)[35]. We found that, when F518 is present as in the WT FP of B41, it would clash with VRC34.01 CDRs and abrogate binding (Supplementary Fig. 7d). Replacing Val for Phe at position 518 in the FP B41mut1 variant restores the geometry of the epitope and prevents a clash. Thus, the nature of the residue at position 518, but not 519, is critical for VRC34.01-like antibody engagement; it is also potentially relevant to HIV-1 escape from neutralization as Phe is present in ~2% of group M isolates.

We detected no binding of PGT151 and ACS202 to either the B41 SOSIP.664 trimer or its mut1 variant (Supplementary Fig. 8), although both bNAbs are known to bind the BG505 SOSIP.664 FP[34,50]. PGT151 neutralizes the B41 Env-pseudotyped virus, but is not able to bind the B41 SOSIP.664 trimer[28,50,51]. A T538F mutation to the FPPR stabilizes this flexible epitope at the gp120/gp41 interface of the B41 SOSIP.664 trimer and enhances PGT151 binding[46]. The major difference between VRC34.01, vFP16, vFP20, and other FP-directed antibodies is the presence of a hydrophobic YYYY motif in CDRH3 of PGT151[44] that interacts with the FP and is predictably similar in ACS202[34]. However, VRC34.01, vFP16, and vFP20 bind to soluble BG505 SOSIP despite not having this YYYY motif in their CDRH3. Thus, we can categorize these FP-directed antibodies into two separate classes: the first, which includes ACS202, is dependent on a hydrophobic patch in CDRH3 for interaction while, in contrast, VRC34.01-like antibodies, lack this requirement. This variety in angles of approach results from multiple FP conformations that also enable residues to be shared at this epitope, but in completely different recognition modes (Supplementary Fig. 9). Notably, ACS202 had slightly stronger binding to its autologous AMC011 SOSIP.v4.2 trimer, compared to BG505 SOSIP.664, despite an identical FP; there may therefore be some (albeit minor) dependence on position 229 (K229 in BG505 but N229 in AMC011), while glutamate remains identical at positions 83 and 87[34]. In AMC011, an E87A change abrogates ACS202 binding completely. In HIV-1 group M, glutamate is at position 87 in ~55% of isolates, and glycine in ~15%. We deduce that G87 in B41, as found in clone 2D7 of AMC011 (consensus of the infectious molecular clones 2D6, 2D7, and 2G9 of the individual viruses from month 8 that elicited the ACS202 lineage), hampers ACS202 binding and aids viral escape[34]. Despite sharing partially overlapping epitopes, the binding stoichiometries of these FP-directed bNAbs on the trimer is distinct; ACS202, VRC34.01, vFP16, and vFP20 bind in a stoichiometry of three Fabs per trimer, while only 2 PGT151 Fabs bind per trimer due to an antibody-induced asymmetry in the trimer[44].

Our analysis suggests that FP flexibility allows for different modes of antibody recognition of the epitopes in which the FP is

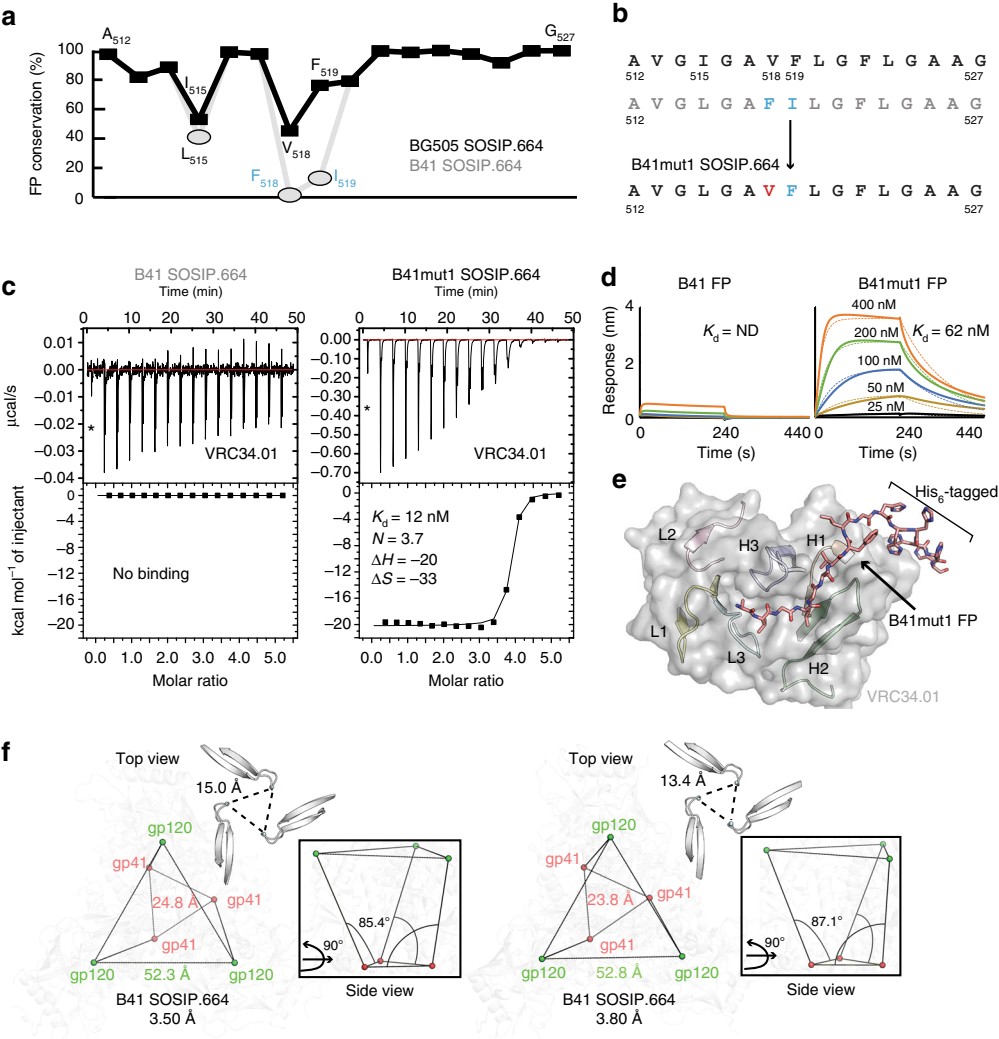

**Fig. 3** Restoration of the VRC34.01 epitope on B41 and delineation of Env breathing. **a** Amino acid conservation of the FP (X-axis: residues 512–527, Y-axis: percent conservation) in BG505 SOSIP.664 (black) and B41 SOSIP.664 (gray). The difference in conservation is shown in rectangles for BG505 SOSIP.664 and ovals for B41 SOSIP.664. For assessing conservation of FP residues, 5451 HIV-1 sequences from Los Alamos National Laboratory (LANL) HIV-1 database were analyzed. **b** The FP residues of both isolates are aligned to highlight the differences at positions 515, 518, and 519. Construction of the B41mut1 SOSIP.664 is illustrated in the lower panel. **c** No binding is observed between FP-directed Fab VRC34.01 with wild-type FP of B41 SOSIP.664 (left panel). Restoration of VRC34.01 binding to the mutated B41mut1 SOSIP.664 FP ($K_d$ = 12 nM, three Fabs bind per trimer) is illustrated (right panel). The enthalpy and entropy are measured in kcal per mol and cal per mol per deg, respectively. Data points not included in the fit are indicated by an asterisk. All binding experiments are measured by ITC and reported values are averages from two independent measurements. **d** Binding kinetics of B41 FP and FP (mut1) peptide variants of VRC34.01, as determined by BLI. **e** Crystal structure of His-tagged B41mut1FP peptide bound to VRC34.01. **f** View down the three-fold trimer apex of B41 SOSIP.664 showing opening and closing of the trimer at 3.50 Å (left) and 3.80 Å (right) resolutions, respectively. Inter-V2 distances (Å; dashed lines) are measured between the Cα residues of R166 (cyan). Top views of both crystal structures showing center of mass of each protomer (gp120: green, gp41: red) overlaid on the Env cartoon (light gray) for orientation. Inter-gp41 and inter-gp120 distances (measured in Å; lines) are measured between the center of masses. The insets of the two B41 structures are shown in the side view (right). The angle is measured between center of mass of gp41 and gp120 of a protomer to reflect the movement between domains that leads to a slight opening of the trimer

involved. This same flexibility also creates challenges for epitope stabilization, which is critical for effective germline antibody engagement and vaccine design when targeting this region of the virus.

**gp41 plays a role in Env breathing.** Conformational changes induce opening of soluble Env in response to receptor and/or co-receptor binding[31]. However, structural movements, such as relaxation of variable regions and movement between and within sub-domains, induce partial opening/closing of the trimer termed breathing. Previous studies have shown that CD4-binding opens Env,

and that some bNAbs can also induce trimer opening[21,31–33,40,41,52]. To understand the structural implications of induced opening versus breathing in the same isolate, we compared both structures of the B41–Fab complexes. We observed breathing at the trimer apex when identical antibodies were bound to B41 SOSIP.664 (Fig. 3f and Supplementary Fig. 10). We next examined the cause of apical opening. On measuring the angle between the center of masses of Cα atoms of gp120 and gp41 sub-domains (either excluding or including the FP) of the two structures, we detected a small outward tilt (~1° and ~0.5°, respectively) in the gp120 subunits of the B41 SOSIP.664 trimer in the 3.5 Å structure compared to the 3.8 Å structure (Fig. 3f) relative to gp41. Although small, this tilt angle is sufficient to induce

an ~1.6 Å opening at the trimer apex. We also observed movement within both sub-domains and twisting of the trimer (Supplementary Movies 1 and 2). However, rearrangements within gp41 alone are not enough to trigger full trimer opening when compared to that induced by receptor (CD4) binding. Thus, from these results, we observe that conformational isomerism and changes/rearrangements in the gp41 FP and FPPR can have distal effects on the trimer apex.

**Comparison of open and closed states of prefusion B41.** We next evaluated conformational rearrangements that occur in B41 trimers upon binding of CD4. We compared B41 SOSIP.664 bound by b12 [CD4-binding site antibody, PDB 5VN8[31]] and by sCD4 and Fab 17b [CD4-induced antibody, PDB 5VN3[31]], as observed by cryo-EM, and compared them with the 3.5 Å crystal structure of the closed B41 trimers bound by PGT124 and 35O22. The overall Cα r.m.s.d between the open and closed B41 protomer/trimer structures varies from 2.6 Å/21.7 Å (5VN8) to 8.3 Å/

18.7 Å (5VN3). This divergence is a reflection of the changes in conformation of both subunits, as well as relative movements between sub-domains and rotations within the trimer (Supplementary Fig. 11). The b12-bound B41 trimer shows a large outward movement of gp120 (22.7 Å between the center of masses of the closed structure and 5VN8, with their gp41s aligned), that is sufficient to open up the trimer apex. In the sCD4-bound (5VN3) structure, the apical opening (21 Å, calculated as described above) results from more localized V1/V2 rearrangements compared to B41$_{PGT124+35O22}$ (Fig. 4a, d and Supplementary Movie 3). In the closed prefusion B41$_{PGT124+35O22}$ structure, the FP is largely solvent exposed (and stabilized by crystal contacts) and extends away from the trimer core, whereas the FPPR faces inward as seen in all soluble Env trimers (Fig. 4e). The engagement of CD4 induces major movements in gp120 relative to gp41, thus forming a new pocket to accommodate the FP in a conformation that points towards the trimer core, while the FPPR is dislodged from

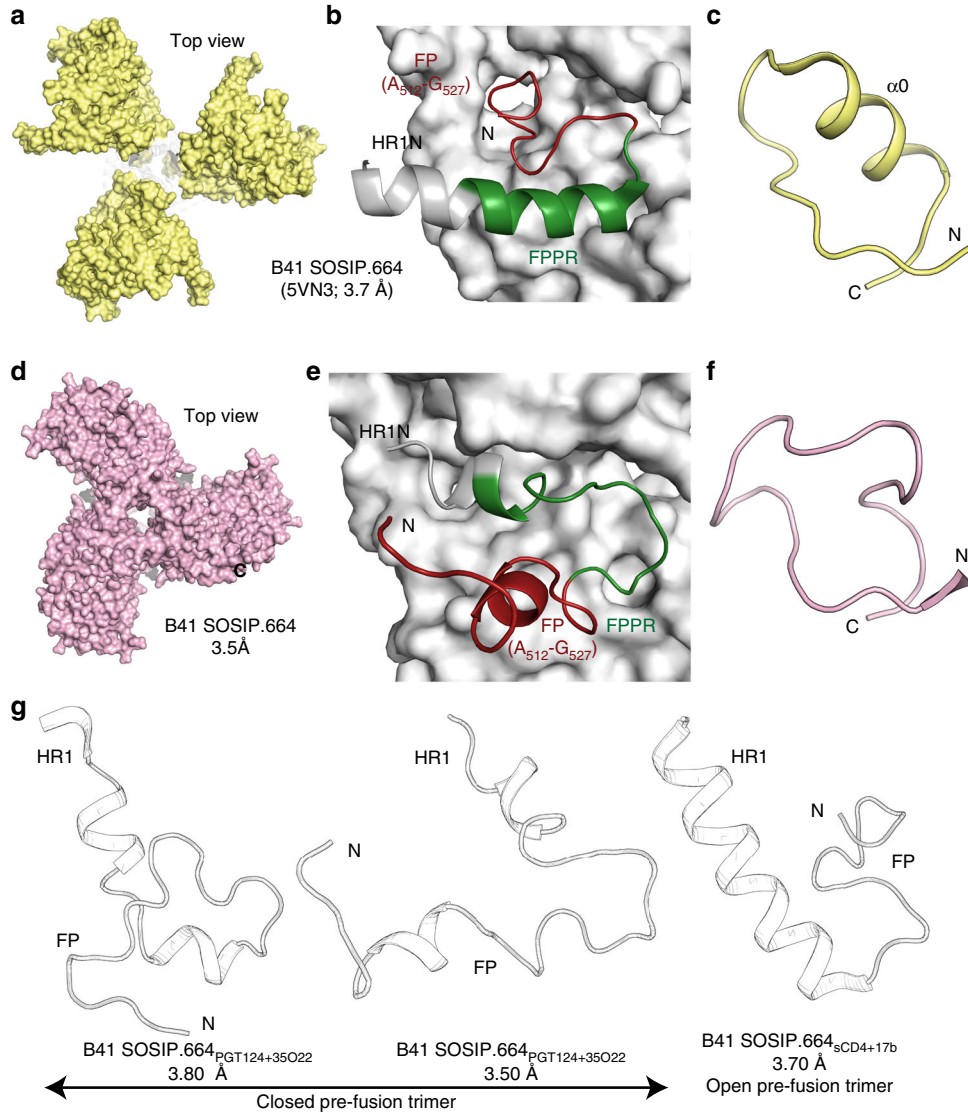

**Fig. 4** Comparison of open and closed prefusion structures of B41 SOSIP.664. **a** Top view of open B41 SOSIP.664 bound to sCD4 and 17b cryo-EM structure (PDB 5VN3). **b** Binding of sCD4 and Fab17b induce conformational changes in the FP and its proximal region of gp41. The FP (red) is inserted towards the trimer core and the FPPR (green) extends away from trimer interface. **c** Binding of sCD4 to B41 SOSIP.664 propagates structural changes to the C1 region that drive the formation of the α0 helix (residues 63–72). **d** Top view of B41 SOSIP.664 bound to Fabs PGT124 and 35O22 shows the closed prefusion Env structure. **e** The free N-terminus of the FP (red) in closed prefusion structure of B41 SOSIP.664 points away from the trimer core compared to the open prefusion trimer. **f** The closed prefusion B41 SOSIP.664 trimer lacks the well-formed α0 helix shown in **c**. **g** Conformational flexibility of B41 SOSIP.664 FP in closed and open prefusion trimers

its usual closed pre-fusion position (Fig. 4b). The closed prefusion B41 structure also lacks the helical α0 (residues 63–72) secondary structure in the C1 region that is observed in the open B41$_{sCD4}$ $_{+17b}$ structure (Fig. 4c, f)[31], while this region is disordered in B41$_{b12}$, indicative of high local flexibility. Our results suggest that the transformation in the FP and FPPR gp41 sub-domains regulates the transition between partially open and fully closed prefusion conformational states of the SOSIP.664 trimer. This inherent conformational flexibility of the FP (Fig. 4g) illustrates that various distinct stable conformational geometries and locations of the FP region can be observed for the prefusion state of Env. These findings highlight and help to explain the differential recognition modes by different FP-directed antibodies[37].

## Discussion

For structure-guided HIV-1 Env vaccine design, it is important to critically analyze how the conservation, location, and conformational flexibility of the FP influence not only its functional role, but also its recognition by bNAbs and ensuing viral escape pathways. Furthermore, the notion of the FP being poorly immunogenic or not accessible in the HIV-1 Env prefusion state has been countered by recent studies where the free N-terminal residues of FP on the Env trimer have been shown to interact with bNAbs PGT151, ACS202, VRC34.01, vFP16, and vFP20[34–36,50,51]. In contrast to the solvent-exposed FP, the N-terminal FP residues of the B41 SOSIP.664 trimer bound to sCD4 and 17b, relocate to a shallow pocket in the trimer interface[31]. This alternative location for the HIV-1 Env FP is more reminiscent of the buried location of its counterpart in the cleaved influenza hemagglutinin[53,54]. Conversely, FP residues 512–519 are exposed on the cleavage-independent BG505 NFL.664 trimer even when no FP-directed antibodies are bound[19], which is similar to what is found for the uncleaved human hemagglutinin FP[55–57]. Multiple crystal structures with 35O22 (gp120/gp41 interface antibody) show similar FP/FPPR conformations[13,43,58,59], as also seen in BG505 NFL.664[19], DS-SOSIP[18] and the PGT-151-unbound protomer in JRFLΔCT[44] that do not have 35O22 bound. These observations reflect that 35O22 does not preselect FP conformations or influence the conformational dynamics of the FP epitope.

In this study, we draw on two high-resolution crystal structures of a soluble, clade B HIV-1 Env trimer, B41 SOSIP.664 to illustrate the dynamic nature of the FP that arises from substantial conformational variation in the FPPR. We were also able to restore the B41 SOSIP.664 epitope for VRC34.01, a bnAb that neutralizes 49% of 208 HIV-1 strains. This epitope restoration could engage a subset of HIV-1 group M strains that possibly escape from VRC34.01-like antibodies due to a steric clash caused by a mutation to a large hydrophobic residue at position 518. Our structure not only sheds light on FP flexibility and conformation when unbound, but also highlights the importance of the highly (~98.5%) conserved F522 residue that acts as the first pivot allowing FP structural plasticity. The structure also reaffirms the highly conserved M530 residue as a key anchor for the prefusion state of gp41. M530 is locked inside a deep cavity formed by the very highly conserved tryptophan clasp[13], perhaps to avoid premature conformational rearrangements in gp41 before receptor engagement that could disrupt membrane fusion with the target host cell. In addition, an M530A mutant virus lacked fusogenic activity[60]. It is plausible that some of the potential energy required for fusion is locked within the tryptophan clasp during the folding of the trimer into its prefusion state. During receptor engagement, the trimer undergoes a cascade of conformational changes that trigger rearrangements of the FP and FPPR; the resulting release of steric constraints allow gp41 to transition from its metastable state to a highly stable six-helix bundle. The

multi-stage conformational rearrangements must also release energy (by displacing F522 and possibly M530 from their binding pockets) that helps drive the insertion of the FP into the host cell membrane. Thus, for virus–cell fusion to occur, these processes need to be triggered by receptor and co-receptor binding and must not be initiated prematurely.

Finally, our crystallographic structures of the closed prefusion B41 SOSIP.664 trimer illustrate the structural plasticity of the FP and the modes of engagement of different FP-directed bNAbs from varying angles. The large conformational changes in the FP and the rearrangement in the FPPR may also provide opportunities for design or selection of small molecules to block the conformational rearrangements associated with membrane fusion. Further, redesign of the B41 FP to restore binding for the VRC34.01-class of bNAbs should engage the population of HIV-1 isolates that disrupt antibody binding at this epitope critical for viral fusion via an escape mutation. Thus, these insights provide further clues for specific targeting of the FP epitope in HIV-1 Env in structure-based immunogen design.

## Methods

**Protein expression, purification, and complex formation.** B41 SOSIP.664 trimers were expressed in FreeStyle HEK 293F (Invitrogen Catalog number:R79007) and 293S (ATCC CRL-3022) cells. The B41mut1 SOSIP.664 FP double mutant (with a deletion of Ile519 nucleotide codon and insertion of Val nucleotide codon between residues A517 and F518 in the original sequence) was generated using a polymerase chain reaction-based mutagenesis approach and expressed in FreeStyle HEK 293S cells. Secreted proteins were purified with a 2G12-coupled affinity matrix followed by size exclusion chromatography (SEC) on a Superdex 200 16/600 column (GE Healthcare) (Supplementary Fig. 1c). B41 SOSIP.664 and B41mut1 SOSIP.664 trimers were analyzed with SDS–PAGE. Fabs of bNAbs PGT124[42], 35O22[61], ACS202[34], VRC34.01[35], and PGT151[50] were produced by transient transfection of FreeStyle HEK 293F cells and purified using either a λ or κ Capture select column (GE Healthcare) followed by ion-exchange chromatography (GE Healthcare) and, if required, by further SEC on a Superdex 75 16/600 column.

**Crystallization and data collection.** Many combinations of Fab-SOSIP.664 trimer complexes (in a 3.5:1 molar ratio of Fab:trimer), with and without deglycosylation using endoH digestion (New England Biolabs), were subjected to crystallization trials. The glycans protected by antibody interaction were preserved, while those accessible to endoH glycosidase were trimmed to the chitobiose core. The SEC-purified B41 SOSIP.664 trimer complexes were concentrated to ~10 mg/ml before being screened at both 4 and 20 °C using our high-throughput CrystalMation$^{TM}$ robotic system (Rigaku) at TSRI[62]. High-quality crystals of PGT124 and 35O22 bound to B41 SOSIP.664 trimers formed in two different conditions: 0.1 M HEPES (pH = 7.0), 8% PEG8000; and 0.1 M calcium acetate, 0.1 M MES (pH = 6.0), 15% PEG400. Crystals were harvested and cryoprotected with 25% glycerol for the first condition and 10% PEG400 for the second condition, followed by immediate flash cooling in liquid nitrogen. Peptides corresponding to the wild type and mutant (mut1) FP from B41 SOSIP.664 trimers were ordered from Peptide 2.0 Inc. (>98% purity). The Fab VRC34.01-B41mut1FP complex was crystallized at ~10 mg/ml (TBS) with an Fab:peptide molar ratio of 1:5. Crystals formed in 0.2 M sodium sulfate, 20%(w/v) PEG3350 and were cryoprotected with 10% ethylene glycol before plunging in liquid nitrogen. Data were collected at Advanced Photon Source (APS) 23-IDD and 23-IDB and Stanford Synchrotron Radiation Lightsource (SSRL) 12-2 beamlines.

**Structure determination and refinement.** The B41 SOSIP.664 trimer-Fab complex crystals from the above two conditions diffracted to 3.50 Å and to 3.80 Å resolution and the data were indexed and integrated using HKL2000[63] in P23 and P6$_3$ space groups, respectively. The structures were solved by molecular replacement using Phaser[64] with Fab structures (PDB 4R26[42] for PGT124, and PDB 4TOY[61] for 35O22) and Env BG505.664 gp140 (PDB 5CEZ[45]) as search models. The P6$_3$ complex was refined to $R_{cryst}/R_{free}$ of 29.4%/31.1% with 99.9% completeness and unit cell parameters $a = b = 129.3$ Å, $c = 313.0$ Å, and the P23 crystal structure to $R_{cryst}/R_{free}$ of 30.4%/32.1% with 100% completeness and unit cell parameters $a = b = c = 212.3$ Å (Supplementary Table 1). The crystal of B41mut1FP bound to VRC34.01 Fab diffracted to 1.97 Å and the diffraction data were indexed in the P2$_1$ space group. Fab VRC34.01 (PDB ID: 5I8C[35]) was used for molecular replacement. Model building was carried out with Coot and refinement was carried out with Phenix[65,66]. The structure was refined to $R_{cryst}/R_{free}$ of 19.9%/23.8% with 99.6% completeness and unit cell parameters $a = 42.4$ Å, $b = 123.6$ Å, $c = 101.2$ Å, $\beta = 90.8°$ (Supplementary Table 2). Structure quality was determined by MolProbity. For the Fabs, residues were numbered according to the Kabat numbering scheme[67], and Env was numbered according to the HXB2 system[68].

Structure validation was performed using the PDB Validation Server (validate. wwpdb.org), PDB-care[69] and Privateer[70]. Data refinement statistics are shown in Supplementary Tables 1 and 2.

**Isothermal titration calorimetry**. ITC experiments were performed with a MicroCal Auto-iTC200 instrument (GE Healthcare). All proteins were purified and dialyzed overnight in a buffer containing 20 mM Tris and 150 mM NaCl (pH 7.4). Ligands, i.e., the FP-interacting bNAbs PGT151, VRC34.01 and ACS202, were present in the syringe at concentrations ranging between 90 and 150 μM. Substrates, i.e., B41 SOSIP.664 and B41mut1 SOSIP.664 trimers, were placed in the cell at concentrations ranging between 5.0 and 6.0 μM. Protein concentrations were measured by UV absorbance at 280 nm using extinction coefficients. Binding experiments were carried out with the following parameters: cell at 25 °C, 16 injections of 2.5 μl each, injection interval of 180 s, injection duration of 5 s, and reference power of 5 μcals. The Origin 7.0 software was used to fit and integrate the titration peaks to a single-site-binding model, and dissociation constants ($K_d$), molar reaction enthalpy ($\Delta H$), and stoichiometry of binding ($N$) were calculated thereafter.

**Biolayer interferometry**. Binding assays were performed using an Octet Red instrument (FortéBio). FPs of either B41 WT or B41mut1 (His-tagged at the C-terminus) were loaded on to Ni-NTA biosensors at a concentration between ~10 and 50 μg/ml in kinetics buffer (1PBS, pH7.4, 0.01%BSA and 0.002% Tween 20). The binding kinetics for the association of the VRC34.01 Fab were measured in a dilution series between 400 and 25 nM. The data were fit with a 1:1 global fitting model to estimate the $K_d$.

**Differential scanning calorimetry (DSC)**. The thermal stability of B41 SOSIP.664 trimers, either unbound or in complex with Fabs PGT124 and 35O22, in TBS buffer from 20 to 120 °C was measured using a MicroCal VP-Capillary calorimeter (Malvern) at a scanning rate of 90 °C/h. Data were analyzed using the VP-Capillary DSC automated data analysis software. The resulting data were fit to a non-two-state model.

**Analysis of Env trimer breathing**. To determine breathing of the B41 SOSIP.664 trimer, we measured the inter-V2 distances between Cα atoms of several residues at different locations in the V2 loop in both structures. We calculated the center of mass using the Cα atoms of each gp120 and gp41 sub-domain protomer separately, either including or excluding the FP, using the Center of Mass function in PyMol. We then measured the distance between the centers of mass of the gp120 and gp41 sub-domains. Finally, we measured the angle between gp120 and gp41 of each protomer in an Env trimer that reflected tilting of a protomer. This tilting reflects breathing or partial opening of trimer apex. The centers of masses (COMs) of the two crystal structures are different indicating distinct gp41 conformations. This difference is reflected in the distances measured between the COMs (Fig. 3f). The distance between COMs does not reflect the opening or closing of the trimer because centers of mass lie within respective sub-domains and do not factor in tilting between the sub-domains.

**Determination of FP sequence conservation**. We inspected 5451 HIV-1 strain sequences from Los Alamos HIV-1 database (http://www.hiv.lanl.gov/) to determine the frequency with which different residues are present in the FP. All HIV-1 sequences were aligned using Jalview.

**Reporting summary**. Further information on experimental design is available in the Nature Research Reporting Summary linked to this article.

## Data availability

The coordinates and structure factors reported in this manuscript have been deposited in the Protein Data Bank (PDB) with accession codes 6MCO, 6MDT, and 6ME1. The authors declare that all other data supporting the findings of this study are within this article and its Supplementary Information files, or are available from the authors upon request.

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

## Acknowledgements

We thank Y. Hua, H. Tien, and Drs. X. Dai, X. Zhu, M. Elsliger, and N. Tzarum for excellent technical help; Dr. M.J. van Gils for sharing the expression vectors for ACS202. Diffraction data were collected at the Advanced Photon Source (APS) beamline 23-IDD, 23-IDB, and Stanford Synchrotron Radiation Lightsource (SSRL) beamline 12-2. Use of the APS was supported by the DOE, Basic Energy Sciences, Office of Science, under contract no. DE-AC02-06CH11357. Use of the SSRL was supported by the US Department of Energy, Basic Energy Sciences, Office of Science, under contract no. DE-AC02-76SF00515. This work was supported by the HIV Vaccine Research and Design (HIV-RAD) program (P01 AI110657) (R.W.S., J.P.M., I.A.W.), the International AIDS Vaccine Initiative Neutralizing Antibody Center and CAVD, and the Center for HIV/AIDS Vaccine Immunology and Immunogen Discovery (CHAVI-ID UM1 AI100663) (I.A.W., A.B.W.).

## Author contributions

Project design by S.K., A.S., and I.A.W. B41 SOSIP.664 construct design and characterization by P.P., R.W.S. and J.P.M. Protein expression and purification by S.K. and A.S. X-ray crystallography by S.K.; ITC, BLI, and DSC by S.K. Data analysis and interpretation by S.K., A.S., and I.A.W. Manuscript written by S.K., A.S., J.P.M., and I.A.W. and edited by S.K., A.S., R.W.S., J.P.M., A.B.W., and I.A.W. This is publication number 29737 from The Scripps Research Institute.

## Additional information

**Competing interests:** The authors declare no competing interests.

