## [Peer Review File · Nature Communications]

Reviewers' Comments:

Reviewer #1:

Remarks to the Author:

Kumar et al. reports three crystal structures of Fab/Env trimer complexes based on B41, expanding the HIV Env structural database and highlighting the most ordered views of the crucially important fusion peptide. Crystallographic results are complemented by appropriate biochemical/binding studies. The scope of the presented studies is extensive, and the results are fairly carefully interpreted. Experimental details are sparse, but adequate to draw the conclusion that experiments have been performed with adequate rigor - though the PDB validation reports were needed to support these conclusions. The presentation thoroughly extracts all possible impact from the results in a clear and very readable format.

Overall, the results merit publication, representing an important extension of prior studies. However, the overall conclusions regarding fusion peptide structure/dynamics, while valid and biologically relevant, are somewhat colored by the limited resolution (and resulting lower quality of the refinements) and effects of crystal contacts. Accompanying biochemical studies do help flesh out the results, though.

In general, the manuscript is acceptable for publication as submitted, though electron density maps should be replaced with ones calculated with less biased coefficients than straight Fo-Fc's.

Reviewer #2:

Remarks to the Author:

The manuscript by Kumar et al addresses the structural flexibility of the N-terminal fusion peptide (FP) of Env gp41 that plays an important role in virus-cell entry via its ability to insert in the host cell membrane. The authors report three new crystal structures, two of a SOSIP trimer bound to antibodies 35O22 and PGT124 that crystallized in two different space groups, and a third of a fusion peptide mutant bound to VRC34.01 Fab. The authors use structural information to restore the binding of VRC34.01 to B41 SOSIP by mutating residues Phe 518 to Val. The crystal structure of VRC34.01 bound to the fusion peptide from B41 with the F518V mutation provides atomic level information for the resistance of B41 to VRC34.01.

One concern is that the major point made in this paper, that the fusion peptide is very flexible, is a finding that was previously published (Xu et al., NatMed, 2018; Dingens et al., Plos Pathogens 2018). This paper adds to the evidence on structural plasticity of the fusion peptide, which is useful although not novel. Another concern is that often this paper fails to distinguish between findings that are new to this paper and those that were already known. For example, the insight that FP residue Phe 522 acts as an anchor or a pivot for the flexible fusion peptide, was published and discussed at length in the Dingens et al. Plos Pathogens paper. Thus, the structural data and analyses presented in this paper are interesting, though not novel, and therefore, the findings reported here are more suited to a more specialized structure journal. The paper will require major revisions to improve text quality and content and in its current version is not suitable for publication. The following lists some of the key points to be addressed.

1. The authors have used mixed reference formatting – “number’ and ‘first author’ (lines 62-63).
2. Line 100, ‘these findings help uncover the structural flexibility in this region...’ is not a new finding and has been described previously.
3. Line 153-155. The peptide is caught in a crystal contact and that likely compromises the

conformation of the free peptide.

4. Line 167-171. The flexibility of the FP has been reported in an earlier publication. Dingens et al reported that "Phe 522 anchors the flexible fusion peptide via van der Waals interactions to $\alpha 6$ helix of gp41, and $\beta 5$ and $\beta 7$ strands of gp120".

5. Line 217. Not a novel conclusion.

6. Line 336-337. The report of FP flexibility, conformation and the importance of the conserved F22 were all conclusions in the Dingens et al., 2018 paper.

7. Supplemental Fig 1. The symmetry of the SOSIP-Fab bound peak indicates that complex dissociation and thermal denaturation are tightly coupled. The author should include the DSC denaturation profile of the Fab alone in Suppl. Fig 1b. The heat capacity C_p for the SOSIP protein alone is also noted to be markedly higher than that of the Fab-bound complex. The mw of the standards should be marked in the inset showing SDS-PAGE results in c and d.

8. The residues that comprise the FPPR should be explicitly stated in the manuscript.

9. Line 84 – the authors should include a citation for the recent SOSIP DEER experiments (Stadtmueller et al 2018, Immunity) suggesting significant motion in gp41.

10. Several claims in the paper are unsupported. For example, lines 173-176 state that loss of F522 interface interaction and associated conformational change enables the trimer to acquire multiple conformations. This is not supported by any observation in the manuscript, nor does the provided reference (27) support this claim. If this is a hypothesis based upon the data and literature that the authors would like to put forward, it must be made clear to the reader.

11. The hypothesis and how the analysis will help to support it in lines 196-199 is not very clear. This sentence should be rephrased.

12. The measurements made in PyMol are not convincing. Using a single V2 residue's alpha carbon, which lies in a loop, does not provide a sufficient measure of the openness at the apex. Loop movement alone could account for the change despite the neighboring apex residues remaining in largely the same position. In order to make this measurement robust, several or all residues that constitute the apex should be used to calculate changes in apex opening. The center of mass calculations are also problematic. It is not clear whether the FP was included in the gp41 COM calculations. Considering the fairly minor COM shifts observed, inclusion of the conformationally variable FP could have a major impact on the result. In this particular case it is also recommended that sidechains be excluded in the COM calculations.

Reviewer #3:

Remarks to the Author:

Kumar et al. have determined two crystal structures of the B41 SOSIP Env in complex with the PGT124 and 35022 bnAbs, which target the V3-glycan supersite and the gp120-gp41 bridging regions, respectively. The two crystal structures determined revealed different conformations of the fusion peptide and fusion peptide proximal region of the gp41 subunit of HIV Env. Based on structural data, the authors further modified the FP region to recapitulate binding to the VRC34.01 bnAb, which could not bind to the wild-type B41 Env.

It's interesting to see that two conformations were captured crystallographically for the exact same complex. The structural data presented here would be of interest to others in the field as there is limited atomic-level data on the different steps of viral fusion. Specifically, the data presented illustrate some of the dynamic properties of the gp41 region, which could be informative for the viral-host fusion process as well as immunogen design. Their design of a peptide mutant that could engage a bnAb, which otherwise could not bind to the wild-type version, further illustrates the applicability to immunogen design strategies.

Even so, there are some questions that arise based on the techniques that were used and some of the assumptions that were made. These are outlined below along with some suggestions, which should be addressed to strengthen the conclusions being made.

Major comments:

Crystal packing can influence the conformation of a protein complex. Is it possible that the different FP and FPPR conformations visualized can be influenced by crystal packing? Cryo-EM structures would confirm these conformations and potentially identify additional conformations and therefore strengthen the conclusions that are being made.

Line 72. Are these soluble trimers really native? They have their cytoplasmic tails truncated and there have been reports that antigenicity can be different for such Envs (see J. Chen et al. Science 2015). Could working with a full length Env impact the conformational dynamics of the gp41 region compared to what is reported here? If so, how?

In the discussion on Env "breathing": How does one know that the gp41 conformations influence the gp120 conformations and not vice versa? Also, how does one know the differences observed aren't due to crystal artifacts?

The Env trimer structures determined were in complex with a bnAb that can only bind the trimer – 35o22. Could this impact what conformations of the FP and FPPR could be observed? The implications of this should be included in the discussion.

Other comments:

Figures:

Figure 3c. While it is apparent that there is increased binding for the B41mut1 SOSIP compared to the wild-type, the plot of integrated heats below the raw data for the mutant doesn't seem to match up with the size of the peaks obtained. For example, peaks 2-10 look like they decrease in size, but the curve below does not indicate this. Is this the correct plot or are the peaks getting wider in the later injections?

Supplementary Fig. 3. It's difficult to see the crystal contacts to determine whether or not they have an effect on the conformation of Envs that were determined. It would be helpful to have a zoomed in view in addition to what is currently provided, i.e. with the gp41 and gp120 subunits along with their crystal contacts clearly visible from both crystal structures obtained.

Minor points and questions:

Line 332 – It should say "epitope for VRC34.01, a bnAb ..."

Line 395 – There's a typo in Lighsource

How were the short fusion peptides produced?

We thank the reviewers for their careful consideration, positive feed-back, and detailed comments on our work. Please find our point-by-point response to the reviewers' comments and changes in the revised manuscript (green).

Reviewers' comments:

Reviewer #1 (Remarks to the Authors)

Kumar et al. reports three crystal structures of Fab/Env trimer complexes based on B41, expanding the HIV Env structural database and highlighting the most ordered views of the crucially important fusion peptide. Crystallographic results are complemented by appropriate biochemical/binding studies. The scope of the presented studies is extensive, and the results are fairly carefully interpreted.

We thank the reviewer and appreciate the positive feedback on our manuscript.

Experimental details are sparse, but adequate to draw the conclusion that experiments have been performed with adequate rigor - though the PDB validation reports were needed to support these conclusions.

We are pleased that the PDB validation reports adequately support our conclusions and satisfy the reviewer. We have also added DSC experiments and additional structural analysis as suggested by Reviewer 2.

The presentation thoroughly extracts all possible impact from the results in a clear and very readable format. Overall, the results merit publication, representing an important extension of prior studies. However, the overall conclusions regarding fusion peptide structure/dynamics, while valid and biologically relevant, are somewhat colored by the limited resolution (and resulting lower quality of the refinements) and effects of crystal contacts. Accompanying biochemical studies do help flesh out the results, though.

Response: We thank the reviewer for the supportive comments. We agree that the structures of both of our soluble HIV-1 Env complexes, based on B41 SOSIP.664 trimers bound to the PGT124 and 35O22 Fabs, are of lower resolution (3.50 Å and 3.80 Å) compared to the B41mut1 fusion peptide (FP) bound to VRC34.01 Fab (1.97 Å). The lower resolution occurs in all but one (PDB 5CEZ¹, resolution 3 Å) of the soluble HIV-1 Env structures and arises from anisotropy in the diffraction and probably also from some heterogeneity in the dense glycan shield on Env together with the flexibility of the surface-exposed hypervariable loops. However, the deposited PDB structures have good Ramachandran and $R_{\text{free}}/R_{\text{cryst}}$ statistics for the resolution reported. We also supplement our structural conclusions with biochemical and biophysical experiments, as the reviewer noted, to provide relevant complementary and supporting information.

In general, the manuscript is acceptable for publication as submitted, though electron density maps should be replaced with ones calculated with less biased coefficients than straight $F_o - F_c$'s.

Response: We have updated all electron density maps in the manuscript with composite omit maps to remove model bias and address the reviewer's comment.

Reviewer #2 (Remarks to the Author):

The manuscript by Kumar et al addresses the structural flexibility of the N-terminal fusion peptide (FP) of Env gp41 that plays an important role in virus-cell entry via its ability to insert in the host cell membrane. The authors report three new crystal structures, two of a SOSIP trimer bound to antibodies 35O22 and PGT124 that crystallized in two different space groups, and a third of a fusion peptide mutant bound to VRC34.01 Fab. The authors use structural information to restore the binding of VRC34.01 to B41 SOSIP by mutating residues Phe 518 to Val. The crystal structure of VRC34.01 bound to the fusion peptide from B41 with the F518V mutation provides atomic level information for the resistance of B41 to VRC34.01.

One concern is that the major point made in this paper, that the fusion peptide is very flexible, is a finding that was previously published (Xu et al., NatMed, 2018; Dingens et al., Plos Pathogens 2018). This paper adds to the evidence on structural plasticity of the fusion peptide, which is useful although not novel. Another concern is that often this paper fails to distinguish between findings that are new to this paper and those that were already known. For example, the insight that FP residue Phe 522 acts as an anchor or a pivot for the flexible fusion peptide, was published and discussed at length in the Dingens et al. Plos Pathogens paper.

Thus, the structural data and analyses presented in this paper are interesting, though not novel, and therefore, the findings reported here are more suited to a more specialized structure journal. The paper will require major revisions to improve text quality and content and in its current version is not suitable for publication.

Response: We do not agree that our results lack novelty, and/or have largely been reported previously. Our reading of the papers by Xu et al., Nat Med 2018 and by Dingens et al., PLoS Pathogens 2018 leads us to differ from the perspective above of what these reports contain. These papers describe structural and/or biochemical data for anti-FP antibodies *when bound to soluble BG505 Env*, information that is then used to describe the conformational flexibility of the FP epitope in its *bound state*. Neither paper reports on the structural flexibility of the FP epitope in the context of the *unliganded* native-like soluble trimer, which is the focus of our work here. We therefore believe that the information that we are presenting is of substantial value for the design and interpretation of trimer-based immunization studies that are intended to elicit anti-FP antibodies that can neutralize the virus. While the structural plasticity of the FP can be inferred from the various bound conformations, it has not been demonstrated explicitly whether this arises because of conformational isomerism in the unbound FP or via an induced fit-mechanism that applies to the FP upon antibody binding.

Our results also provide an explanation for the role of the F522 anchor and have implications for understanding the conformational changes in the soluble B41 SOSIP trimer immunogen when residue F522 becomes freed spontaneously from its anchored location. We believe that is another new finding that is beneficial to immunogen design. In contrast, Dingens et al. (PloS Pathogens 2018) conclude from their anti-FP antibody-bound structures that F522 is the FP anchor point. They also hypothesize that mutations at residue G524 mediate viral escape either because they alter how the epitope is presented or because they change the conformational dynamics of the FP itself.

Taken together, our manuscript describes independent information on the FP that does not merely reiterate what is already known but instead adds a new dimension to our understanding of the structure and function of the FP, particularly from the HIV immunogen design perspective. Hence, we believe that our results are worthy of publication in Nature Communications.

Please find below our point-by-point response to the reviewer's questions.

The following lists some of the key points to be addressed.

1. The authors have used mixed reference formatting – “number” and “first author” (lines 62-63).

Response: We thank the reviewer for noticing this formatting error, which we have corrected in the revised manuscript.

2. Line 100, “these findings help uncover the structural flexibility in this region...” is not a new finding and has been described previously.

Response: We have discussed this difference in the interpretation of our results above. Our structures show the structural flexibility of the unbound FP epitope as opposed to previously published results showing flexibility in FP-bound conformations^{2,3,4}. Our results also describe large rearrangements in the FPPR when the FP is in its unbound state in soluble cleaved HIV Env, which has not been described to date. We have now modified the statement on lines 101-102 in the manuscript to be more explicit: ‘*Our results map the structural flexibility in this FP region when the epitope is unliganded and not bound by FP-specific bnAbs.*’

3. Line 153-155. The peptide is caught in a crystal contact and that likely compromises the conformation of the free peptide.

We thank the reviewer for commenting on this aspect. In most crystal structures of soluble Env trimers described to date without an anti-FP antibody, for example PDB IDs: 5CEZ, 4TVP, 5FYK, 5FYJ, 5UM8, 6B0N, the FP is involved in crystal contacts that help stabilize this hydrophobic region. Each of these FP structures has a similar directionality (pointing downwards) and conformation to our B41 SOSIP.664 structure in the P6₃ space group, although not necessarily with identical contacts. Our second structure in a different space group indicates the range of conformations that the FP can access in its unliganded form. Furthermore, the large displacement of the FPPR has not been observed in any of the previous prefusion soluble trimers (cleaved or cleavage-independent), with or without anti-FP antibodies or crystal contacts. These findings support our proposal that these different conformations of the flexible FP are accessible in solution and are not seen primarily as a result of crystal contacts.

4. Line 167-171. The flexibility of the FP has been reported in an earlier publication. Dingens et al reported that “Phe 522 anchors the flexible fusion peptide via van der Waals interactions to $\alpha 6$ helix of gp41, and $\beta 5$ and $\beta 7$ strands of gp120”.

Response: We agree and reference Dingens et al, PLoS Pathogens 2018, while mentioning that F522 is a pivot for the FP. However, reiterating our previous reply, we emphasize that the above publication does not discuss the flexibility of the unbound FP.

5. Line 217. Not a novel conclusion.

Response: Xu et al., Nat Med 2018, report that, in the soluble BG505 DS-SOSIP Env background, the binding of VRC34.01 is *partially* sensitive to the identity of residue 518. However, the B41 wild-type FP has Phe at position 518. Our structural and binding data (as illustrated in Figure 3c and Supplementary Fig 7d) clearly identify F518 to be the principal factor that abrogates VRC34.01 binding. Given the different FP compositions between BG505 and B41, this is important information for designing Env immunogens intended to induce anti-FP antibodies and hence we feel that this is a new finding.

6. Line 336-337. The report of FP flexibility, conformation and the importance of the conserved F22 were all conclusions in the Dingens et al., 2018 paper.

Response: Although Dingens et al., PLoS Pathogens 2018, observe F522 to be an anchor for the FP, the description of the flexibility is based on the multiple conformations of the anti-FP bound structures. Here we describe multiple conformations acquired by a flexible unbound FP

within the same BG505 Env protein, which provide important information for vaccine design targeting this epitope.

7. Supplemental Fig 1. The symmetry of the SOSIP-Fab bound peak indicates that complex dissociation and thermal denaturation are tightly coupled. The author should include the DSC denaturation profile of the Fab alone in Suppl. Fig 1b. The heat capacity Cp for the SOSIP protein alone is also noted to be markedly higher than that of the Fab-bound complex. The mw of the standards should be marked in the inset showing SDS-PAGE results in c and d.

Response: We thank the reviewer for this excellent suggestion. We now include the DSC profiles for unbound Fabs PGT124 and 35O22 and have labeled the SDS-PAGE with molecular weights in supplementary Fig 1 and modified the figure legend and main text accordingly.

8. The residues that comprise the FPPR should be explicitly stated in the manuscript.

Response: We have now included the numbering of FPPR residues at the point of first occurrence.

9. Line 84 – the authors should include a citation for the recent SOSIP DEER experiments (Stadtmueller et al 2018, Immunity) suggesting significant motion in gp41.

Response: We agree, and have added the citation.

10. Several claims in the paper are unsupported. For example, lines 173-176 state that loss of F522 interface interaction and associated conformational change enables the trimer to acquire multiple conformations. This is not supported by any observation in the manuscript, nor does the provided reference (27) support this claim. If this is a hypothesis based upon the data and literature that the authors would like to put forward, it must be made clear to the reader.

Response: We have rephrased this sentence (lines 177-181) to ‘Such flexibility in the FP may aid the Env trimer in accessing the conformations observed in the pre-fusion state, antibody-bound states (e.g. PGT151), and the transition to intermediate states. These include the fully closed native-like state and those in equilibrium between the closed and more open conformations observed by NS-EM in different strains and clades.’ This should make it clear that we are extrapolating from our data and from what is reported in Pugach et al, 2015 (in the manuscript), to make what we think is a reasonable argument. We are not sure what the reviewer means by “several claims in this paper are unsupported” as only one such claim is actually identified.

11. The hypothesis and how the analysis will help to support it in lines 196-199 is not very clear. This sentence should be rephrased.

Response: This section of results describes the testing of our hypothesis that the registry of the FP residues is important for anti-FP bNAb engagement. This hypothesis stems from our analysis of the FP region using >5400 sequences in the LANL database. We use our structural observations to guide the design of mutations that restore the VRC34.01 epitope on the B41 SOSIP.664 trimer. We have now rephrased the sentence on lines 201-205 to ‘To test our hypothesis that the amino acid sequence and register of certain FP residues have an important role in antibody binding, we analyzed the FP sequence conservation in HIV-1 group M, including recombinants, and compared the sequences to BG505 SOSIP.664 (Fig. 3a).’

12. The measurements made in PyMol are not convincing. Using a single V2 residue’s alpha carbon, which lies in a loop, does not provide a sufficient measure of the openness at the apex. Loop movement alone could account for the change despite the neighboring apex residues remaining in largely the same position. In order to make this measurement robust, several or all residues that constitute the apex should be used to calculate changes in apex opening. The

center of mass calculations are also problematic. It is not clear whether the FP was included in the gp41 COM calculations. Considering the fairly minor COM shifts observed, inclusion of the conformationally variable FP could have a major impact on the result. In this particular case it is also recommended that sidechains be excluded in the COM calculations.

Response: We thank the reviewer for this comment. We have now calculated inter-V2 distances of several C α residues located at the interior, base and tip of the V2 loops to reflect the breathing at the trimer apex (included in a new supplementary Fig. 10) and modified the main text accordingly. For COM calculations, we have now used only C α atoms for both gp120 and gp41 sub-domains and recalculated COM for the gp41 sub-domain, with separate calculations that either include or exclude the FP (updated Figure 3). The new calculations reinforce our observations between the two identical trimer-antibody complexes in different space groups that can be attributed to breathing.

Reviewer #3 (Remarks to the Authors)

Kumar et al. have determined two crystal structures of the B41 SOSIP Env in complex with the PGT124 and 35022 bnAbs, which target the V3-glycan supersite and the gp120-gp41 bridging regions, respectively. The two crystal structures determined revealed different conformations of the fusion peptide and fusion peptide proximal region of the gp41 subunit of HIV Env. Based on structural data, the authors further modified the FP region to recapitulate binding to the VRC34.01 bnAb, which could not bind to the wild-type B41 Env. It's interesting to see that two conformations were captured crystallographically for the exact same complex. The structural data presented here would be of interest to others in the field as there is limited atomic-level data on the different steps of viral fusion. Specifically, the data presented illustrate some of the dynamic properties of the gp41 region, which could be informative for the viral-host fusion process as well as immunogen design. Their design of a peptide mutant that could engage a bnAb, which otherwise could not bind to the wild-type version, further illustrates the applicability to immunogen design strategies.

Even so, there are some questions that arise based on the techniques that were used and some of the assumptions that were made. These are outlined below along with some suggestions, which should be addressed to strengthen the conclusions being made.

Response: We thank the reviewer for the positive comments on the implications of our results for informing immunogen design by providing new knowledge of the structural conformations available to the fusion machinery. Please find our point-by-point responses to the reviewer's comments.

Major comments:

1) Crystal packing can influence the conformation of a protein complex. Is it possible that the different FP and FPPR conformations visualized can be influenced by crystal packing?

Response: We thank the reviewer for raising this critical question. From the multiple crystallographic structures (e.g., PDB IDs: 5CEZ, 4TVP, 5FYJ, 5FYK) published of the soluble HIV Env SOSIPs without anti-FP antibodies bound, the cleaved FP usually stacks against the hydrophobic face of 35O22 (a gp120/gp41 interface, non-FP antibody) from a symmetry mate to sequester itself from solvent exposure via crystal packing. This sequestering of the FP has also been observed in the cryo-EM structure (where crystal contacts do not apply) of B41 SOSIP.664 trimers in the prefusion and CD4-induced intermediate conformations⁵. Recently, crystal contacts between the FP and an N332-directed bNAb PGT122 have been reported to occur in the structure of a cleavage-independent soluble BG505 NFL Env⁶, which in fact has its

unbound-FP in a highly similar conformation to the VRC34.01-bound FP. Additionally, the cryo-EM structure of PGT151-unbound FP of JRFL wild type Δ CT (PDB 5FUU) also seems to follow a similar trend in directionality. Irrespective of the design platform or genotype, the directionality (pointing downwards) of the FP conformation remains similar in all aforementioned crystal structures previously published, as we find in our B41 structure in the $P6_3$ space group, but not in the $P23$ space group. Also, large rearrangements observed in the FPPR in our study cannot be readily attributed to crystal packing. Taken together, the evidence implies that the two distinct structures that we have observed for the same Env-antibody complex (B41 SOSIP.664 with PGT124 and 35O22) represent bona fide alternative structural conformations and are not merely crystal packing artifacts.

Cryo-EM structures would confirm these conformations and potentially identify additional conformations and therefore strengthen the conclusions that are being made.

Response: We thank the reviewer for this suggestion and we have used cryo-EM extensively in our studies of HIV Env. However, similar technical challenges affect both cryo-EM and crystallography when determining atomic level resolution structures of highly flexible sub-regions of a protein complex as evidenced in our previous studies with HIV Env^{6,7}. High resolution within flexible regions of macromolecular complexes can usually only be achieved after stabilization by epitope-specific antibodies, which is not our objective in this study.

Line 72. Are these soluble trimers really native? They have their cytoplasmic tails truncated and there have been reports that antigenicity can be different for such Envs (see J. Chen et al. Science 2015). Could working with a full length Env impact the conformational dynamics of the gp41 region compared to what is reported here? If so, how?

Response: We thank the reviewer for raising this question. This has been a long-standing question that has been systematically answered in the field^{8,9}. The soluble immunogens mimic the HIV virion-associated Env in antigenicity^{10, 11, 12, 13, 14}, bNAb recognition^{15, 16}, conformational dynamics¹⁷, structure^{1, 6, 18, 19, 20}, design^{13, 21, 22, 23} and post-translational modifications, particularly glycosylation²⁴. Structures of trimeric Envs containing the transmembrane region but not the cytoplasmic tail have been described that do not show major deviations from the soluble structures. The presence of the cytoplasmic tail has little impact on membrane fusion as found by Chen et al., Science 2015, and may or may not have pronounced effects on the FP dynamics. However, we concur that this has to remain as ‘speculation’ until a full-length structure with the cytoplasmic domain at high enough atomic resolution is solved and is outside the scope of this study.

In the discussion on Env “breathing”: How does one know that the gp41 conformations influence the gp120 conformations and not vice versa? Also, how does one know the differences observed aren’t due to crystal artifacts?

Response: There have been multiple recent reports of structural heterogeneity in the gp41 subdomains^{5, 17} that are likely sources of Env metastability²¹. In addition to these biochemical studies, multiple crystal and cryo-EM structures have found more disorder to be present in conserved regions of gp41 than in gp120^{18, 19, 20}. Together, these various observations made using a range of techniques all suggest larger movements occur within the gp41 subdomain than in gp120. Our new structures are therefore entirely consistent with a prior body of literature, implying that they are not crystal artifacts, as now mentioned on lines 184-187. “The current arsenal of crystal and cryo-EM structures illustrate that the FP has a significant dynamic range of conformations, independent of crystal contact formation, which facilitate bNAb (VRC34.01, ACS202, vFP16, vFP20 and PGT151) engagement from various angles of approach.”

The Env trimer structures determined were in complex with a bnAb that can only bind the trimer – 35o22. Could this impact what conformations of the FP and FPPR could be observed? The implications of this should be included in the discussion.

Response: Previously published crystallographic structures with the gp120/gp41 interface binding 35O22 antibody have been found with similar FP/FPPR conformations ^{18, 25, 26, 27}. However, the BG505 NFL.664 ⁶, DS-SOSIP ⁴ and JRFL Δ CT ⁷ (PGT151-unbound protomer) structures, none of which involves a 35O22 complex, also reveal FP conformations that resemble the 35O22-bound structures. This implies that the presence of 35O22 does not influence the conformational dynamics of the FP. The distinct FP/FPPR conformations of the identical B41 SOSIP.664+PGT124+35O22 complex presented in this manuscript thus implies that the gp120/gp41 interface binding antibody 35O22 does not preselect these FP conformations as now mentioned on lines 335-339. “ Multiple crystal structures with 35O22 (gp120/gp41 interface antibody) show similar FP/FPPR conformations ^{13, 43, 58, 59}, as also seen in BG505 NFL.664 ¹⁹, DS-SOSIP ¹⁸ and the PGT-151-unbound protomer in JRFL Δ CT ⁴⁴ that do not have 35O22 bound. These observations reflect that 35O22 does not preselect FP conformations or influence the conformational dynamics of the FP epitope.”

Other comments:

Figures:

Figure 3c. While it is apparent that there is increased binding for the B41mut1 SOSIP compared to the wild-type, the plot of integrated heats below the raw data for the mutant doesn't seem to match up with the size of the peaks obtained. For example, peaks 2-10 look like they decrease in size, but the curve below does not indicate this. Is this the correct plot or are the peaks getting wider in the later injections?

Response: In Figure 3c, the raw data are plotted in μ cal/sec against time. The processed data are the normalized integration of each injection plotted against the molar ratio. Both raw and processed data are linked via the X-axis, so that the integrated area for each peak appears directly below the corresponding peak in the raw data. The integrated area of each peak that indicates change in enthalpy (heat change) for injections 2 to 10 are minuscule ($\Delta H = -7.36$ to -7.67) on the Y-axis of the final figure. Gradually shortening but widened peak profiles (shown in the raw data) when the system begins to approach saturation ($\Delta H = -5.5$ to 0.1) is reflected in formation of a sigmoidal curve as seen in the final figure.

Supplementary Fig. 3. It's difficult to see the crystal contacts to determine whether or not they have an effect on the conformation of Envs that were determined. It would be helpful to have a zoomed in view in addition to what is currently provided, i.e. with the gp41 and gp120 subunits along with their crystal contacts clearly visible from both crystal structures obtained.

Response: We thank the reviewer for this comment. We have now incorporated a zoomed-in panel in Supplementary figure 3.

Minor points and questions:

Line 332 – It should say “epitope for VRC34.01, a bnAb ...”

Response: We corrected the error and appreciate being made aware of it.

Line 395 – There's a typo in Lighsource

Response: The typo has been corrected - thanks.

How were the short fusion peptides produced?

Response: The C-terminal His-tagged peptides were ordered from Peptide 2.0 Inc. with >98% purity, as stated in Methods (under crystallization and data collection subheading).

References:

1. Garces F, *et al.* Affinity maturation of a potent family of HIV antibodies is primarily focused on accommodating or avoiding glycans. *Immunity* **43**, 1053-1063 (2015).
2. Xu K, *et al.* Epitope-based vaccine design yields fusion peptide-directed antibodies that neutralize diverse strains of HIV-1. *Nat Med* **24**, 857-867 (2018).
3. Dingens AS, *et al.* Complete functional mapping of infection- and vaccine-elicited antibodies against the fusion peptide of HIV. *PLoS Pathog* **14**, e1007159 (2018).
4. Kong R, *et al.* Fusion peptide of HIV-1 as a site of vulnerability to neutralizing antibody. *Science* **352**, 828-833 (2016).
5. Ozorowski G, *et al.* Open and closed structures reveal allostery and pliability in the HIV-1 envelope spike. *Nature* **547**, 360-363 (2017).
6. Sarkar A, *et al.* Structure of a cleavage-independent HIV Env recapitulates the glycoprotein architecture of the native cleaved trimer. *Nat Commun* **9**, 1956 (2018).
7. Lee JH, Ozorowski G, Ward AB. Cryo-EM structure of a native, fully glycosylated, cleaved HIV-1 envelope trimer. *Science* **351**, 1043-1048 (2016).
8. Ward AB, Wilson IA. The HIV-1 envelope glycoprotein structure: nailing down a moving target. *Immunol Rev* **275**, 21-32 (2017).
9. Sanders RW, Moore JP. Native-like Env trimers as a platform for HIV-1 vaccine design. *Immunol Rev* **275**, 161-182 (2017).
10. Derking R, *et al.* Comprehensive antigenic map of a cleaved soluble HIV-1 envelope trimer. *PLoS Pathog* **11**, e1004767 (2015).
11. Sanders RW, *et al.* A next-generation cleaved, soluble HIV-1 Env trimer, BG505 SOSIP.664 gp140, expresses multiple epitopes for broadly neutralizing but not non-neutralizing antibodies. *PLoS Pathog* **9**, e1003618 (2013).
12. Julien JP, *et al.* Asymmetric recognition of the HIV-1 trimer by broadly neutralizing antibody PG9. *Proc Natl Acad Sci U S A* **110**, 4351-4356 (2013).
13. Sanders RW, *et al.* Stabilization of the soluble, cleaved, trimeric form of the envelope glycoprotein complex of human immunodeficiency virus type 1. *J Virol* **76**, 8875-8889 (2002).
14. Binley JM, *et al.* A recombinant human immunodeficiency virus type 1 envelope glycoprotein complex stabilized by an intermolecular disulfide bond between the gp120

- and gp41 subunits is an antigenic mimic of the trimeric virion-associated structure. *J Virol* **74**, 627-643 (2000).
15. Sok D, Burton DR. Recent progress in broadly neutralizing antibodies to HIV. *Nat Immunol* **19**, 1179-1188 (2018).
 16. Kwong PD, Mascola JR. HIV-1 Vaccines Based on Antibody Identification, B Cell Ontogeny, and Epitope Structure. *Immunity* **48**, 855-871 (2018).
 17. Stadtmueller BM, *et al.* DEER spectroscopy measurements reveal multiple conformations of HIV-1 SOSIP envelopes that show similarities with envelopes on native virions. *Immunity* **49**, 235-246 e234 (2018).
 18. Pancera M, *et al.* Structure and immune recognition of trimeric pre-fusion HIV-1 Env. *Nature* **514**, 455-461 (2014).
 19. Lyumkis D, *et al.* Cryo-EM structure of a fully glycosylated soluble cleaved HIV-1 envelope trimer. *Science* **342**, 1484-1490 (2013).
 20. Julien JP, *et al.* Crystal structure of a soluble cleaved HIV-1 envelope trimer. *Science* **342**, 1477-1483 (2013).
 21. He L, *et al.* HIV-1 vaccine design through minimizing envelope metastability. *Sci Adv* **4**, eaau6769 (2018).
 22. Kong L, *et al.* Uncleaved prefusion-optimized gp140 trimers derived from analysis of HIV-1 envelope metastability. *Nat Commun* **7**, 12040 (2016).
 23. Sharma SK, *et al.* Cleavage-independent HIV-1 Env trimers engineered as soluble native spike mimetics for vaccine design. *Cell Rep* **11**, 539-550 (2015).
 24. Struwe WB, *et al.* Site-specific glycosylation of virion-derived HIV-1 Env Is mimicked by a soluble trimeric immunogen. *Cell Rep* **24**, 1958-1966 e1955 (2018).
 25. Barnes CO, *et al.* Structural characterization of a highly-potent V3-glycan broadly neutralizing antibody bound to natively-glycosylated HIV-1 envelope. *Nat Commun* **9**, 1251 (2018).
 26. Stewart-Jones GB, *et al.* Trimeric HIV-1-Env structures define glycan shields from clades A, B, and G. *Cell* **165**, 813-826 (2016).
 27. Pancera M, *et al.* Crystal structures of trimeric HIV envelope with entry inhibitors BMS-378806 and BMS-626529. *Nat Chem Biol* **13**, 1115-1122 (2017).

Reviewers' Comments:

Reviewer #2:

Remarks to the Author:

The authors have revised the manuscript to address the concerns, have provided additional data and answers to the questions raised.

Reviewer #3:

Remarks to the Author:

The manuscript by Kumar et al. provides important new insights into different conformational states of the gp41 fusion peptide (FP) region of the B41 SOSIP.664 Env trimer, critical for the fusion of viral and host membranes. More significantly, their data highlights the conformations of FP in the absence of directly bound ligands, unlike what has been previously published. They provide comparisons to known structures to indicate that the two conformations of FP presented are not due to crystal contacts. They also use their structural data to illustrate that the FP region could be modified to bind to an otherwise non-binding antibody, important for immunogen design strategies. They complement their structural analyses with appropriate binding experiments and clearly present the data.

In summary, all comments have been satisfactorily addressed and their results merit publication.

Reviewers' comments:

Reviewer #2 (Remarks to the Author)

The authors have revised the manuscript to address the concerns, have provided additional data and answers to the questions raised.

Response: We thank the reviewer for the positive remarks.

Reviewer #3 (Remarks to the Author):

The manuscript by Kumar et al. provides important new insights into different conformational states of the gp41 fusion peptide (FP) region of the B41 SOSIP.664 Env trimer, critical for the fusion of viral and host membranes. More significantly, their data highlights the conformations of FP in the absence of directly bound ligands, unlike what has been previously published. They provide comparisons to known structures to indicate that the two conformations of FP presented are not due to crystal contacts. They also use their structural data to illustrate that the FP region could be modified to bind to an otherwise non-binding antibody, important for immunogen design strategies. They complement their structural analyses with appropriate binding experiments and clearly present the data.

In summary, all comments have been satisfactorily addressed and their results merit publication.

Response: We thank the reviewer for the positive comments.